# Soluble ST2 Receptor: Biomarker of Left Ventricular Impairment and Functional Status in Patients with Inflammatory Cardiomyopathy

**DOI:** 10.3390/cells11030414

**Published:** 2022-01-25

**Authors:** Danilo Momira Obradovic, Petra Büttner, Karl-Philipp Rommel, Stephan Blazek, Goran Loncar, Stephan von Haehling, Maximilian von Roeder, Christian Lücke, Matthias Gutberlet, Holger Thiele, Philipp Lurz, Christian Besler

**Affiliations:** 1Department of Cardiology, Heart Center Leipzig at the University of Leipzig, Strümpellstraße 39, 04289 Leipzig, Germany; petra.buettner@medizin.uni-leipzig.de (P.B.); Karl_Ph_Rommel@web.de (K.-P.R.); Stephan.Blazek@medizin.uni-leipzig.de (S.B.); Maximilian.Roeder@helios-gesundheit.de (M.v.R.); holger.thiele@medizin.uni-leipzig.de (H.T.); Philipp.Lurz@medizin.uni-leipzig.de (P.L.); Christian.Besler@medizin.uni-leipzig.de (C.B.); 2Institute for Cardiovascular Diseases “Dedinje”, Faculty of Medicine, University of Belgrade, 11040 Belgrade, Serbia; loncar_goran@yahoo.com; 3Department of Cardiology and Pneumology, Heart Center, University of Göttingen Medical Center, 37099 Gottingen, Germany; stephan.von.haehling@med.uni-goettingen.de; 4German Center for Cardiovascular Research (DZHK), Partner Site Göttingen, 37099 Gottingen, Germany; 5Department of Diagnostic and Interventional Radiology, Heart Center Leipzig, 04289 Leipzig, Germany; Christian.Luecke@helios-gesundheit.de (C.L.); Matthias.Gutberlet@helios-gesundheit.de (M.G.)

**Keywords:** inflammatory cardiomyopathy, endomyocardial biopsy, dilated cardiomyopathy, ELISA, sST2

## Abstract

Introduction: Inflammatory cardiomyopathy (ICM) frequently leads to myocardial fibrosis, resulting in permanent deterioration of the left ventricular function and an unfavorable outcome. Soluble suppression of tumorigenicity 2 receptor (sST2) is a novel marker of inflammation and fibrosis in cardiovascular tissues. sST2 was found to be helpful in predicting adverse outcomes in heart failure patients with reduced ejection fraction. The aim of this study was to determine the association of sST2 plasma levels with cardiac magnetic resonance (CMR) and echocardiography imaging features of left ventricular impairment in ICM patients, as well as to evaluate the applicability of sST2 as a prognosticator of the clinical status in patients suffering from ICM. Methods: We used plasma samples of 89 patients presenting to the Heart Center Leipzig with clinically suspected myocardial inflammation. According to immunohistochemical findings in endomyocardial biopsies (EMB) conducted in the context of patients’ diagnostic work-up, inflammatory cardiomyopathy was diagnosed in 60 patients (ICM group), and dilated cardiomyopathy in 29 patients (DCM group). All patients underwent cardiac catheterization for exclusion of coronary artery disease and CMR imaging on 1.5 or 3 Tesla. sST2 plasma concentration was determined using ELISA. Results: Mean plasma concentration of sST2 in the whole patient cohort was 45.8 ± 26.4 ng/mL (IQR 27.5 ng/mL). In both study groups, patients within the highest quartile of sST2 plasma concentration had a significantly lower left ventricular ejection fraction (LV-EF) compared to patients within the lowest sST2 plasma concentration quartile (26 ± 11% vs. 40 ± 13%, *p* = 0.05 for ICM and 24 ± 13% vs. 51 ± 10%, *p* = 0.004 for DCM). sST2 predicted New York Heart Association (NYHA) class III/IV at 12 months follow-up more efficiently in ICM compared to DCM patients (AUC 0.85 vs. 0.61, *p* = 0.02) and was in these terms superior to NT-proBNP and cardiac troponin T. ICM patients with sST2 plasma concentration higher than 44 ng/mL at baseline had a significantly higher probability of being assigned to NYHA class III/IV at 12 months follow-up (hazard ratio 2.8, 95% confidence interval 1.01–7.6, log rank *p* = 0.05). Conclusion: Plasma sST2 levels in ICM patients reflect the degree of LV functional impairment at hospital admission and predict functional NYHA class at mid-term follow-up. Hence, ST2 may be helpful in the evaluation of disease severity and in the prediction of the clinical status in ICM patients.

## 1. Introduction

Myocardial inflammation remains one of the major concerns in cardiovascular medicine [1]. Immunological mechanisms that underpin myocardial inflammation cause left ventricular (LV) damage, characterized by remodeling and dilatation, which frequently results in inflammatory cardiomyopathy (ICM) [2]. In comparison to dilated cardiomyopathy (DCM), ICM exhibits substantial variation in disease severity and prognosis, ranging from moderate disease, which can be successfully managed with oral neurohormonal blockade, to advanced disease requiring mechanical circulatory support or heart transplantation [3]. Of note, high numbers of patients present with poor functional status (New York Heart Association [NYHA] class III/IV) and frequent hospital visits 1–3 years after ICM diagnosis [4].

Cardiac magnetic resonance (CMR), as a valuable noninvasive diagnostic tool, offers unique myocardial tissue characterization capabilities with the assessment of biventricular regional and global function [5]. However, the sensitivity of CMR is low for cardiomyopathy and very low for arrhythmic presentation forms of myocarditis and should not be used as the sole method to establish a diagnosis in these cases [6].

N-terminal pro B-type natriuretic peptide (NT-pro-BNP) and cardiac troponin T are established cardiac markers in routine diagnostic work-up of heart failure and acute coronary syndrome, but they have only moderate diagnostic strength and applicability in the setting of myocardial inflammation [7]. Thus, new clinical predictors and biomarkers are needed to predict undesirable clinical outcomes in ICM patients [8].

ST2 is a member of the Toll-like/interleukin (IL)-1 receptor superfamily. The ST2 gene, found on chromosome 2q12, is expressed in four isoforms, two of which include a transmembrane receptor (ST2 ligand, ST2L) and a soluble, serum circulating receptor (sST2) [9]. ST2 in vivo binds to IL33, known to have antihypertrophic and antifibrotic effects on cardiomyocytes [10]. In this way, sST2 may serve as an adverse “decoy receptor” for circulating IL-33, minimizing the protective effects of IL33 on the cardiovascular system [10]. Additionally, ST2 signaling is involved in inflammatory processes, particularly, in antigen-presenting cells, type 2 CD4^+^ T-helper cells, and in the production of Th2-associated cytokines [11].

The premise that sST2 mirrors the degree of tissue inflammation and fibrosis on the molecular level qualifies sST2 as a potential biomarker for risk stratification of ICM patients. It may help to identify individuals more prone to adverse outcomes, who could profit from more intensive clinical monitoring and therapy. In this study, we sought to determine whether sST2 reflects the degree of functional and structural LV impairment as quantified by echocardiography in ICM patients, as well as to test whether the baseline sST2 plasma level is suitable to predict NYHA functional status and frequency of adverse cardiovascular events in ICM patients at mid-term follow-up.

## 2. Methods

### 2.1. Study Design

For the purposes of our study, we used plasma samples of 89 patients presenting with clinically suspected myocarditis at the Heart Center Leipzig between August 2012 and May 2015. In all study participants, the signs of persistent myocardial damage were present at hospital admission (electrocardiographic [ECG] abnormalities, elevated enzymes of myocardial injury, LV dysfunction). All patients underwent cardiological work-up including: (1) LV endomyocardial biopsy [EMB], (2) CMR imaging, (3) echocardiography, (4) ECG, (5) laboratory analysis, and (6) clinical assessment with thorough medical history examination. Only patients after comprehensive immunohistochemical/histological assessment of EMB specimens and exclusion of relevant coronary artery disease (defined as stenosis >50% of vessel diameter by angiography) were included in the present analysis.

The study was conducted according to the Declaration of Helsinki and was approved by local ethical entities. All patients provided informed written consent before their participation.

Twelve months after the study enrolment, trial participants were contacted by telephone, and information concerning physical wellbeing, functional NYHA class, cardiovascular or non-cardiovascular serious adverse events was obtained and documented by study personnel, blinded to study objectives.

### 2.2. Endomyocardial Biopsy and Immunohistochemical Analysis

EMB sampling was performed via arterial access using myocardial biopsy forceps (Teleflex Medical Tuttlingen, Tuttlingen, Germany). Using fluoroscopic guidance, multiple EMB samples (five to six) were taken from different locations within the LV. Extensive assessment of myocardial specimens was performed at the Department of Molecular Pathology, University Hospital Tubingen (Tubingen, Germany), as previously published [12,13,14,15,16]. Histological analysis was performed by an independent pathologist according to previously published protocols and criteria [12,13,16,17,18].

For the purposes of immunohistology analysis, tissue sections were processed according to the avidin–biotin immunoperoxidase method (Vectastain Elite Kit; Vector, Burlingame, CA, USA). Subsequently, the monoclonal antibodies for CD3 (T cells; Novocastra Laboratories, Newcastle, UK), CD68 (macrophages; DAKO, Hamburg, Germany), and human leukocyte antigen-(HLA)DR (DAKO) were added according to previously published protocols [16]. Myocardial inflammation was defined as the detection of ≥14 infiltrating immune cells/mm^2^ (CD3-positive T lymphocytes and/or CD68-positive macrophages) in addition to enhanced HLA class II expression in professional antigen-presenting immune cells according to classification criteria of WHO/International Society and Federation of Cardiology Task Force on the Definition and Classification of Cardiomyopathies.

According to immunohistological findings on EMB, patients were divided into two groups, i.e., patients with ICM and patients without immunohistological evidence of myocardial inflammation (DCM) [7].

### 2.3. CMR Imaging

CMR was performed using a 1.5-T scanner (Intera CV, Philips, Best, The Netherlands) or a 3-T scanner (Verio, Siemens Healthcare, Erlangen, Germany). Image analysis was performed with a standard post-processing platform (cmr42, version 5.1.0; Circle Cardiovascular Imaging, Calgary, AB, Canada) by two experienced observers in consensus. Cine, T2w, T1w, and PSIR were realized as multislice stacks, covering the whole LV, whereas mapping was performed using single slices. Myocardial edema was quantified through visual inspection of a whole myocardium at T2-weighted black-blood imaging, as well as by calculating the T2 ratio as previously described [5]. Briefly, a myocardial region and a skeletal muscle region of interest were drawn, whereupon the subsequent signal intensity of the myocardium was divided by the signal intensity of the skeletal muscle. Edema was diagnosed if the T2 ratio was greater than 1.9.

Early gadolinium enhancement ratio (EGE) was calculated as follows. Standardized myocardial and skeletal muscle regions of interest were drawn on one axial section before and after contrast material administration. EGE ratio was considered pathologic when greater than or equal to 4 [5]. The myocardium was visually assessed on late gadolinium-enhanced (LGE) images and considered suspicious for myocarditis in cases of focal signal intensity alterations with a subepicardial or intramyocardial pattern typical of myocarditis [5]. Lake Louise Criteria ([LLC], T2-weighted edema, EGE, and LGE) were deemed positive for myocarditis with the presence of at least two of three LLC [5].

### 2.4. sST2 Assay

Plasma samples were collected from the peripheral cubital vein into EDTA-containing tubes, centrifuged immediately, and then stored at −80 °C for subsequent analysis. Soluble ST2 levels were assessed in baseline samples using a highly sensitive sandwich monoclonal immunoassay in duplicate (Presage ST2 Assay, Critical Diagnostics, New York, NY, USA), with a lower limit of detection of 2 ng/mL, an upper limit of detection of 200 ng/mL, an intraassay coefficient of variation of <4%, and an interassay coefficient of variation of <6.4%.

### 2.5. NT-proBNP, CK-MB, and Troponin T

NT-proBNP plasma levels were assessed using Elecsys proBNP II Immuno-Assay. In brief, 15 µL of patient serum samples was incubated at 20 °C with two monoclonal antibodies (biotinylated and ruthenium-complexed) against epitopes in the N-terminal domain of proBNP (amino acids 1–76 and 1–108). After the addition of streptavidin-coated microparticles, magnetic bead-mediated immune-complex isolation and purification was conducted. The measurements and read-outs were conducted on the MODULAR ANALYTICS E170 test platform. Creatine kinase myocardial band (CK-MB) and high-sensitivity assay cardiac troponin T were analyzed using immunological UV test and ELISA, as previously described [19]. NT-pro BNP, CK-MB, and cardiac troponin T were assessed at a central core lab, blinded to all clinical data.

### 2.6. Statistical Analysis

Continuous variables are presented as mean and standard deviation (SD) or as median and interquartile range, dependent upon the distribution type of the evaluated variables. The Kolmogorov–Smirnov test was used for appraisal of distribution normality. Categorical variables are expressed as frequencies and percentages. Continuous variables were compared between groups using Student’s *t*-test, Mann–Whitney *U*-test, or Kruskal–Wallis one-way analysis of variance, as appropriate. Correlations were evaluated using either Pearson or Spearman correlation tests, including Bonferroni’s corrections. Receiver operating curves (ROC) analysis was applied to evaluate the accuracy of explanatory variables for dichotomous outcome. Youden Index depicted optimal cut-off values from the ROC for prognostic purposes, and areas under the curves (AUCs) were compared by the DeLong method. Logistic regression estimated the odds ratios and 95% confidence intervals (CI) for univariate predictors of the evaluated outcome variables. Kaplan–Meier curves were used to show the cumulative probabilities of the occurrence of the evaluated outcome variables in the patient cohort, accompanied by the log-rank test between the curves. SPSS software version 21.0 (SPSS Inc., Chicago, IL, USA), GraphPad Prism software version 5.04 (GraphPad Software Inc., La Jolla, CA, USA), and MedCalc Statistical Software version 18.6 (MedCalc Software bvba, Ostend, Belgium) were used in the statistical analysis. Statistical testing was conducted on the basis of a 2-sided α = 0.05 significance level assumption.

## 3. Results

The baseline characteristics of the study population are presented in Table 1. According to the immunohistological findings on EMB, 60 patients were assigned to the ICM and 29 patients to the DCM group. Concerning age and body mass index (BMI), there were no significant differences between the evaluated patient groups. In the ICM group, 28% of patients were females, as in the DCM group (24%). The presence of myocardial hyperemia, evaluated as EGE in T1-weighted CMR images, was more frequent in the ICM group (69% vs. 44% of patients, *p* = 0.01). There were no statistically significant differences between the circulatory concentrations of sST2 and NT-proBNP when comparing the ICM and the DCM groups.

In the ICM and DCM groups, we found significant differences in baseline LV-EF between patients within the lowest and the highest sST2 plasma concentration quartiles (40 ± 13% vs. 26 ± 11%, *p* = 0.05 and 51 ± 10% vs. 24 ± 13%, *p* = 0.004, respectively, Figure 1). Concerning the LV end-diastolic volume (LV-EDV), no statistically significant differences across different sST2 plasma concentration quartiles were detected (ICM 174 ± 64 mL vs. 161 ± 66 mL, *p* = 0.71; DCM 194 ± 110 mL vs. 209 ± 63 mL, *p* = 0.87; Figure 2).

Clinical information concerning the evaluated endpoints from follow-up was available for 82 patients (92%). At 12 months follow-up, 40.4% of patients in the ICM group were classified as the NYHA functional class III/IV. A total number of 12 cardiovascular events were documented: two patients died for a cardiovascular reason, four patients experienced malignant arrhythmias requiring implantation of an implantable cardioverter, and six patients were hospitalized at least once for decompensated heart failure. In the DCM group at follow-up, NYHA class III/IV was documented in 28.6% of cases, and adverse cardiovascular events were registered in 29.6% of patients.

Compared to NT-proBNP and troponin T, baseline plasma sST2 exhibited significantly better accuracy in prognostication of NYHA III/IV functional class at follow-up (AUC 0.85 vs. 0.61 vs. 0.57, respectively; Figure 3) in ICM patients. The Youden Index identified a plasma sST2 concentration of 44 ng/mL as the cut-off point on the ROC curve with the optimal prognostic sensitivity/specificity ratio. According to this finding, all ICM and DCM patients were divided into groups, one with a high baseline sST2 level (sST2 concentration higher than 44 ng/mL), and the other with a low sST2 level (lower than 44 ng/mL).

Patients in the ICM group with a high baseline sST2 plasma concentrations showed a significantly higher cumulative probability to present NYHA class III/IV at follow up compared to patients with a lower baseline ST2 levels (HR 2.8, 95% CI 1.01–7.6, *p* = 0.05, Figure 4). In the DCM group, a high plasma sST2 level at baseline was not able to accurately predict the NYHA class at follow-up (HR 0.45, 95% CI 0.1–2.1, *p* = 0.33, Figure 4). A higher sST2 at baseline was not associated with a composite cardiovascular outcome (comprising cardiovascular mortality, malignant arrhythmias, and/or ICD implantation and hospitalization due to heart failure) at 12 months follow-up, irrespective of the study group (HR 3, 95% CI 0.9–1; *p* = 0.08 for the ICM group and HR 0.83, 95% CI 0.18–3.9; *p* = 0.82 for the DCM group, Figure 5).

On univariate analysis including baseline plasma sST2, CMR conformation of ICM (≥2 Lake Louise Criteria), NT-proBNP, and troponin T, only increased sST2 values showed a significant association with the presence of NYHA class III/IV at 12 months follow-up (Table 2).

## 4. Discussion

There is unmet need for reliable and effective prognostic tools in the clinical management of ICM. The plasma levels of sST2 at hospital admission were associated with the degree of functional LV impairment in patients with ICM and DCM. Furthermore, ICM patients with increased baseline plasma levels of sST2 were at a higher risk to present functional NYHA class III/IV at 12 months follow-up, in comparison to ICM patients with low baseline plasma sST2 levels. In these terms, sST2 proved to be a stronger independent predictor compared to NT-pro-BNP and troponin T in the current study population.

Due to the high heterogeneity of the clinical presentation patterns, the plethora of involved molecular and cellular pathophysiological mechanisms, as well as the diversity of potential etiological factors, the clinical management of myocardial inflammation is inevitably based on the application of combined invasive and non-invasive diagnostic modalities. Although widely recognized as the most reliable diagnostic tool, the invasive nature of EMB sampling is accompanied by a small, but permanent risk of life-threatening complications. Thus, in the past years, research efforts in the development of non-invasive diagnostic platforms, primarily based on imaging techniques and biomarker profiling, capable of an accurate diagnostic and prognostic assessment of ICM patients, were intensified.

The introduction of LLC positioned CMR as the imaging technique of choice for patients with clinical suspicion for myocardial inflammation [5]. In a study by Abdel-Aty et al., LLC yielded a sensitivity and specificity of 76% and 96%, respectively, in patients with acute biopsy-proven myocarditis [20]. Nevertheless, due to the variety of patterns and extent of myocardial involvement, an accurate diagnosis may still be missed in a substantial number of patients, especially in those with a chronic course of the disease. Through comprehensive T1 und T2 imaging protocols, Lurz et. al. improved the diagnostic performance of CMR imaging in ICM subgroups of patients with variable duration of symptoms and clinical stadium of myocarditis [21]. However, the current CMR imaging methods fail to predict the course of myocardial inflammation and are not useful in the assessment of clinical outcome, for which reason, additional clinical tools for prognostication in the setting of myocardial inflammation are needed.

A promising approach to risk assessment of ICM patients might include the use of biological markers of specific pathophysiological processes, such as inflammation, fibrosis, and myocyte stretch. Following this approach, the first clinical trials were focused on establishing a link between the plasma concentration of sST2 and the degree of myocardial remodeling after ischemic cardiac damage. Seki et al. showed that IL-33/ST2 signaling in-vivo protects against adverse cardiac remodeling, whereas sST2 as an IL-33 “decoy receptor” may blunt these protective cardiovascular effects [22]. Accordingly, Weinberg et al. detected an inversed correlation between sST2 and LV-EF after acute myocardial ischemia [23].

In line with these previous findings, we observed that higher plasma levels of sST2 were associated with lower values of the LV-EF in ICM and DCM patients. The missing association between sST2 and LVEDD in both evaluated patient cohorts showed that sST2 exhibited a stronger association with functional and then with morphological signs of LV impairment.

The prognostic potential of plasma sST2 levels was first identified in patients with heart failure and acute myocardial infarction. A meta-analysis by Aimo et al., including seven studies with a global population of 6337 patients, found plasma sST2 to be a predictor of both all-cause and cardiovascular event-related death in chronic heart failure outpatients [24]. Shimpo and colleagues observed that in patients with acute myocardial infarction, sST2 levels are higher in those individuals at a higher risk for cardiovascular death and development of congestive heart failure at 30 days [25]. Aside from major adverse cardiovascular events, the sub-analysis of the PRIDE trial firstly established links between clinical features of heart failure and sST2 plasma concentration. Januzzi and collaborators observed a strong association between baseline sST2 levels and mortality at mid-term follow-up in patients presenting to the emergency department with acute dyspnea [26].

Baseline sST2 levels in our ICM cohort reliably predicted functional NYHA status at follow-up and was comparable with NT-proBNP in the prediction of adverse cardiovascular clinical outcomes (death, malignant rhythm disturbances, and/or ICD implantation and hospitalization due to heart failure symptoms). Interestingly, individuals with baseline plasma concentrations of sST2 higher than 44 ng/mL had a significantly higher probability of a worse functional status (NYHA class III/IV) at mid-term follow-up after ICM diagnosis. Noteworthy, in univariate logistic regression, only sST2 proved to be a significant predictor of the functional status at follow-up, whereas positive LLC and baseline plasma levels of NT-pro-BNP and troponin T were not associated with clinical und functional outcomes.

Several blood biomarkers have been studied in the context of myocarditis and inflammatory cardiomyopathy [27]. The levels of troponin I and troponin T can be elevated in cases of myocarditis but lack sensitivity. In the Multicenter Myocarditis Treatment trial, the sensitivity of elevated troponin I levels in patients with biopsy-proven myocarditis was 34%, and the specificity was 89% [28]. NT-proBNP showed neither significant sensitivity nor specificity for myocarditis conformation [29]. Other serum markers of inflammation, including white blood cell count, erythrocyte sedimentation rate, and C-reactive protein levels, can be elevated in acute myocardial inflammation, but show low accuracy for determining the presence of active myocarditis [28]. Although the routine measurement of cardiac biomarkers (Troponin T, NT-proBNP, and C-reactive protein) in patients with suspected myocarditis is frequent in clinical praxis and is thought to be beneficial in terms of diagnosis conformation, currently there are no established myocarditis-specific blood biomarkers that can inform the diagnosis and can determine the presence or absence of active myocardial inflammation.

To our best knowledge, this is the first study evaluating the prognostic performance of sST2 in the setting of ICM. Acknowledging the fact that sST2 is easily/reproducibly measurable and provides insight into the degree of myocardial inflammatory, sST2 may serve as a promising biomarker candidate in the clinical management of ICM patients. In addition, sST2 values are not significantly affected by BMI or by impairment of renal function, which are important cofactors in cardiovascular patients, and are known to limit prognostic performance of other biomarkers such as BNP and NTproBNP.

There are several limitations to be overcome. First, there is scientific evidence that sST2 may have an immunomodulatory role in asthma, sepsis, acute coronary syndrome, pulmonary disease/acute respiratory distress syndrome, and some autoimmune diseases [30,31]. The current study was not adequately powered to exclude cofounding effects of these clinical entities on the study results. Second, the limited number of study subjects further hampers the generalizability of our findings. sST2 measurements in a larger patient population with different demographic, age, sex, and comorbidity profiles could provide additional insights concerning the sST2 distribution pattern across a wider spectrum of clinical entities. Third, at follow-up examination, the compliance of the study participants to the guideline-recommended treatment was not evaluated, which could be a potential cofounding factor for the reported functional status at follow-up and influence NYHA class distribution of the study cohort. Finally, sST2 levels were analyzed in frozen samples, whereas other biomarkers were evaluated in fresh samples. Consequently, there is a risk that the absolute levels of sST2 could have been affected by having been measured in frozen rather than in fresh samples. However, there is evidence that freeze–thaw cycles do not significantly modify sST2 (manufacturer’s disclosure).

## 5. Conclusions

This study shows that the plasma concentration of sST2 is a promising biomarker of functional LV impairment at hospital presentation and NYHA functional status at mid-term follow-up in ICM patients. Further studies are warranted in order to establish the usefulness of sST2 in the risk stratification of patients with ICM.

## Figures and Tables

**Figure 1 cells-11-00414-f001:**
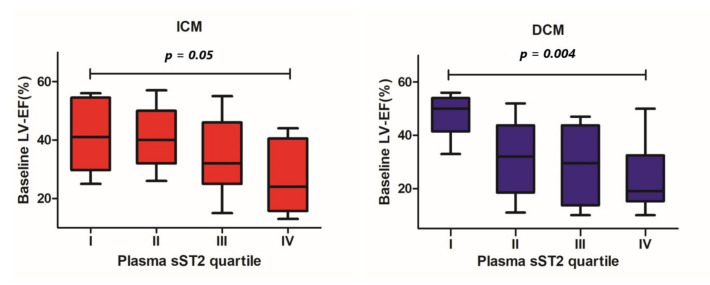
Differences in baseline left ventricular ejection fraction between patients throughout different quartiles of baseline plasma sST2 concentration by patients with ICM and DCM.

**Figure 2 cells-11-00414-f002:**
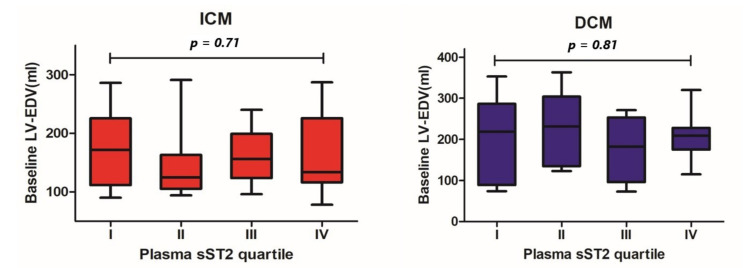
Differences in baseline left ventricular end-diastolic volume between patients throughout different quartiles of baseline plasma sST2 concentration by patients with ICM and DCM.

**Figure 3 cells-11-00414-f003:**
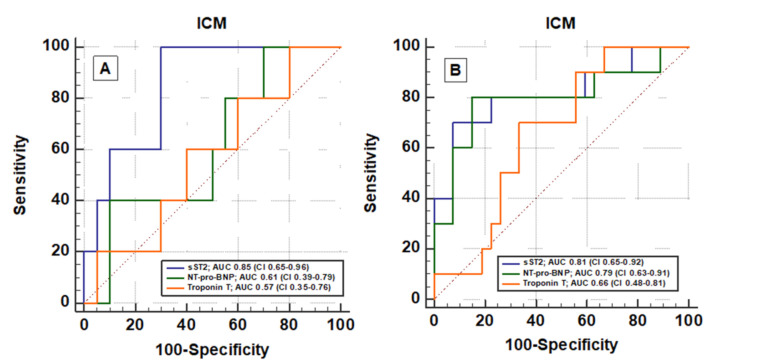
Receiver operating curve (ROC) of sST2, NT-pro-BNP, and Troponin T for the detection of (**A**) NYHA functional class III/IV and (**B**) the presence of adverse cardiovascular events (cardiovascular death, malignant arrhythmias, and/or ICD implantation and hospitalization due to heart failure symptoms) at 12 months follow-up in patients with ICM.

**Figure 4 cells-11-00414-f004:**
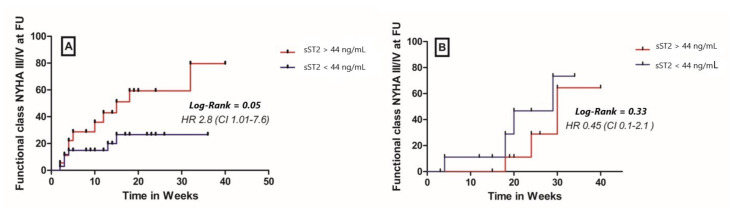
Cumulative probability for assignment to functional class NYHA III/IV at 12 months follow up in (**A**) patients with inflammatory cardiomyopathy (ICM) and (**B**) patients with dilated cardiomyopathy (DCM), FU—follow up.

**Figure 5 cells-11-00414-f005:**
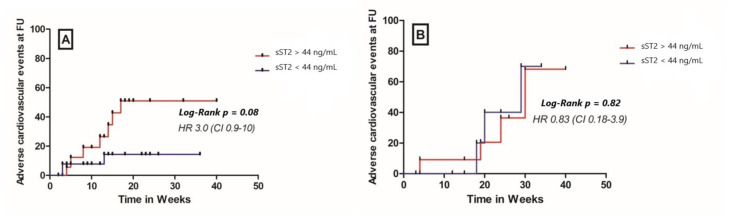
Cumulative probability for composite cardiovascular outcome (cardiovascular death, malignant arrhythmias, and/or ICD implantation and hospitalization due to heart failure) at 12 months follow-up in (**A**) ICM patients and (**B**) DCM patients, FU—follow up.

**Table 1 cells-11-00414-t001:** Baseline characteristics of ICM and DCM patients. BMI—body mass index, CMR—cardiac magnetic resonance, EGE—early gadolinium enhancement, LGE—late gadolinium enhancement, LV-EF—left ventricular ejection fraction, LVEDD—left ventricular end diastolic diameter, LVESD—left ventricular end systolic diameter, sST2—plasma concentration of soluble ST2 receptor, EBV—Epstein–Barr virus; HSV6/HSV7—Human herpes virus 6 and 7.

Clinical Variable	All Patients	Inflammatory Cardiomyopathy (*n* = 60)	Dilated Cardiomyopathy (*n* = 29)	*p* Value
Age (years, mean ± SD)	46 ± 15	47 ± 15	43 ± 14	0.77
Female (%)	26	28	24	0.68
BMI (kg/m^2^)	31	32	31	0.71
NYHA class at presentation (% of patients)				0.85
NYHA class I/II	66	63	71	
NYHA class III/IV	34	37	29	
CMR findings				
T2-weghted edema of LV (% of patients)	11	11	11	0.43
EGE of LV (% of patients)	63	69	44	0.01
LGE of LV (% of LV)	13 ± 8	13 ± 7	12 ± 7	0.88
Echocardiography findings				
LV-EF (%, mean ± SD)	37 ± 16	36 ± 14	35 ± 20	0.69
LVEDD (mm, mean ± SD)	162 ± 70	154 ± 58	180 ± 91	0.76
LVESD (mm, mean ± SD)	106 ± 64	102 ± 54	116 ± 85	0.22
Biomarkers (plasma concentration)				
sST2 (ng/mL; mean ± SD)	45 ± 26	46 ± 29	44 ± 19	0.48
NT-proBNP (pg/mL; mean ± SD)	2866 ± 5010	3208 ± 5726	1994 ± 2255	0.09
CK-MB (U/L; mean ± SD)	64 ± 324	30 ± 35	135 ± 564	0.01
hsCRP (mg/L; mean ± SD)	26 ± 40	29 ± 43	19 ± 32	0.08
Myoglobin (µg/L; mean ± SD)	217 ± 260	198 ± 271	269 ± 228	0.61
Troponin T (ng/L; mean ± SD)	202 ± 539	269 ± 643	64 ± 111	0.009
Immunohistochemistry (left ventricle)				
CD3+ T cells (cells/cm^2^ ± SD)	11 ± 13	14 ± 14	3 ± 2	0.001
CD68+ macrophages (cells/cm^2^ ± SD)	25 ± 19	31 ± 20	11 ± 5	0.009
Enhanced MHC class II expression (n, %)	-	60 (100)	5 (18)	0.001
Viral genome in EMB				
EBV (%)	12	14	-	0.001
HSV6/HSV7 (%)	20	21	-	0.001
Parvovirus B19 (%)	5	4	-	0.001
Enhanced MHC class II expression (%)	64	100	19	0.001
Cardiovascular risk factors				
History of Hypertension (%)	59	56	64	0.48
History of Diabetes (%)	11	11	10	0.97
Tobacco use (%)	42	40	46	0.57

**Table 2 cells-11-00414-t002:** Univariate logistic regression analysis for prognostication of functional capacity and presence of adverse cardiovascular events in patients with inflammatory and dilated cardiomyopathy. sST2—soluble ST2 receptor, LLC—Lake Louise Criteria, ICD—implantable cardioverter defibrillator.

**Functional Class NYHA III/IV at 12 Months**						
	**Inflammatory Cardiomyopathy**			**Dilated Cardiomyopathy**		
**Variables**	**Odds-Ratio**	**CI 95%**	** *p* **	**Odds-Ratio**	**CI 95%**	** *p* **
sST2 (ng/mL)	1.21	1.01–1.30	0.02	0.92	0.80–1.06	0.28
LLC (≥2)	0.92	0.27–3.15	0.90	1.10	0.30–8.79	0.95
NT-pro-BNP (pg/mL)	1.01	1.0–1.11	0.57	1.00	0.99–1.01	0.27
Troponin-T (ng/L)	0.99	0.98–1.01	0.44	0.94	0.93–1.02	0.30
**Cardiovascular****Outcome at 12 Months** (**Cardiovascular** **Mortality,** **Malignant** **Arrhythmias/ICD** **Implantation,** **Hospitalization** **Due** **to** **Heart Failure)**						
	**Inflammatory Cardiomyopathy**			**Dilated Cardiomyopathy**		
**Variables**	**Odds-Ratio**	**CI 95%**	** *p* **	**Odds-Ratio**	**CI 95%**	** *p* **
sST2 (ng/mL)	1.02	0.99–1.10	0.15	1.01	0.92–1.11	0.81
LLC (≥2)	0.81	0.21–3.18	0.76	2.01	0.28–5.54	0.31
NT-pro-BNP (pg/mL)	0.99	0.98–1.01	0.83	1.00	0.99–1.03	0.75
Troponin-T (ng/L)	0.99	0.98–1.01	0.56	0.99	0.95–1.01	0.61

## Data Availability

The data that support the findings of this study are available from the corresponding author on reasonable request.

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
