# Peer review of "Soluble ST2 Receptor: Biomarker of Left Ventricular Impairment and Functional Status in Patients with Inflammatory Cardiomyopathy"

_cells, 2022, doi:10.3390/cells11030414_

Round 1

Reviewer 1 Report

What the manuscript talking about the soluble ST2 receptor as biomarker of heart failure patients is very important and interesting. However, It has been well reported that sST2 are associated with cardiac and inflammatory diseases [Bayes-Genis A, et al. ST2 in heart failure. Circ Heart Fail 2018;11:e005582.] and has established as a cardiac biomarker [Haller PM, et al. Role of Cardiac Biomarkers in Epidemiology and Risk Outcomes. Clin Chem. 2021 67(1):96-106.]. So with the current results, the originality of the work does not merit publication.

Author Response

Thank You for your comments. We also believe that sST2 as a biomarker is well established in population of patients with heart failure (as both references that You kindly stated underline). However, the clinical relevance and importance of sST2 in patients with inflammatory cardiomyopathy, what was primary goal in our study, is not investigated in detail and data concerning its prognostication in this patient collective are scarce.

It is why, we believe that further studies concerning the role of sST2 as a biomarker in inflammatory cardiomyopathy, on even larger patient populations, are warranted and could lead to the improvement in clinical management of these patients.

Reviewer 2 Report

The Authors aimed to study the prognostic value of sST2 in patients suffering from ICM.

The subject is well chosen and addresses a relevant issue. The manuscript reads well. Methods are suitable to answer the question of the study. Results are acceptable. Figures are illustrative and easy to understand. Tables are sometimes overdetailed and long. Discussion and conclusion is appropriate. 

Some minor points need to be addressed in a revised manuscript before considering for publication.

Table 1. Tables should be able to stand alone, I recommend to write ICM and DCM in full as well. 

Figure 2 left figure a technical sign remained present in the manuscript

Author Response

Thank You sincerely for Your review and comments.

In the revised manuscript, we inserted all necessary corrections.

  • Table 1. Tables should be able to stand alone, I recommend to write ICM and DCM in full as well. 

We entered inflammatory cardiomyopathy and dilated cardiomyopathy as you kindly suggested.

  • Figure 2 left figure a technical sign remained present in the manuscript

We removed technical sign form the figure 2.

Reviewer 3 Report

The authors showed that the plasma concentration of sST2 is a usefull biomarker for prediction of LV impairment and NYHA class in ICM patients. The manuscript is quite concise and well written, however, there are some minor typographical errors to be corrected for publication. 

  1. Line 73: effect if IL33 → effects of IL33
  2. Line 195: sST2 und NT-proBNP → sST2 and NT-proBNP
  3. Line 314: due heart failure → due to heart failure
  4. Line 315: 44ng/ml → 44 ng/ml
  5. Line 342: limitations to be acknowledge → limitations to be overcome

Author Response

We are very greatful for Your comments.

All nesseccery corrections are adressed in the revised manuscript and stated belove.

  • Line 73: effect if IL33 → effects of IL33

Apologies for spelling mistake, we corrected it in revised manuscript.

  • Line 195: sST2 und NT-proBNP → sST2 and NT-proBNP

We changed “und” to “and” in revised manuscript.

  • Line 314: due heart failure → due to heart failure

It is corrected.

  • Line 315: 44ng/ml → 44 ng/ml

We inserted space between 44 and ng/ml in corrected version of the manuscript.

  • Line 342: limitations to be acknowledge → limitations to be overcome

We corrected the sentence as suggested in revised manuscript.

Round 2

Reviewer 1 Report

This study is recommended for publication now.